# Vaccination with *Schistosoma mansoni* Cholinesterases Reduces the Parasite Burden and Egg Viability in a Mouse Model of Schistosomiasis

**DOI:** 10.3390/vaccines8020162

**Published:** 2020-04-03

**Authors:** Bemnet A. Tedla, Darren Pickering, Luke Becker, Alex Loukas, Mark S. Pearson

**Affiliations:** Centre for Molecular Therapeutics, Australian Institute of Tropical Health and Medicine, James Cook University, Cairns, QLD 4878, Australia; amarebem6@gmail.com (B.A.T.); darren.pickering@jcu.edu.au (D.P.); luke.becker@jcu.edu.au (L.B.); alex.loukas@jcu.edu.au (A.L.)

**Keywords:** schistosomiasis, vaccine, cholinesterase

## Abstract

Schistosomiasis is a neglected tropical disease caused by parasitic blood flukes of the genus *Schistosoma*, which kills 300,000 people every year in developing countries, and there is no vaccine. Recently, we have shown that cholinesterases (ChEs)—enzymes that regulate neurotransmission—from *Schistosoma mansoni* are expressed on the outer tegument surface and present in the excretory/secretory products of larval schistosomula and adult worms, and are essential for parasite survival in the definitive host, highlighting their utility as potential schistosomiasis vaccine targets. When treated *in vitro* with anti-schistosome cholinesterase (*Sm*ChE) IgG, both schistosomula and adult worms displayed significantly decreased ChE activity, which eventually resulted in parasite death. Vaccination with individual *Sm*ChEs, or a combination of all three *Sm*ChEs, significantly reduced worm burdens in two independent trials compared to controls. Average adult worm numbers and liver egg burdens were significantly decreased for all vaccinated mice across both trials, with values of 29–39% and 13–46%, respectively, except for those vaccinated with *Sm*AChE1 in trial 1. Egg viability, as determined by egg hatching from liver homogenates, was significantly reduced in the groups vaccinated with the *Sm*ChE cocktail (40%) and *Sm*AChE2 (46%). Furthermore, surviving worms from each vaccinated group were significantly stunted and depleted of glycogen stores, compared to controls. These results suggest that *Sm*ChEs could be incorporated into a vaccine against schistosomiasis to reduce the pathology and transmission of this debilitating disease.

## 1. Introduction

Schistosomiasis is caused by an infection with parasitic blood flukes of the genus Schistosoma, killing 300,000 people every year and infecting hundreds of millions more in developing countries [1]. The disease is spread when humans come into contact with the water-borne, infective stage of the parasite—the cercariae—which penetrate the skin and migrate through the vasculature and lungs before entering the venous system, where they become sexually mature adults that pair and mate. Eggs laid by the female are then passed in the urine or feces to the environment, which continues the transmission of schistosomiasis [1]. Eggs are deposited in tissues and organs of the host, such as the liver, and much of the disease pathology is a product of the immune response against these trapped ova, where the ensuing granulomatous lesions lead to fibrosis, which can cause severe circulatory impairment of the affected organs [2].

Despite decades of concentrated research, there is still no effective and practical vaccine against the disease [3]. Furthermore, mass chemotherapy using praziquantel (PZQ)—the only effective anti-schistosomal drug—is complicated by rapid and frequent reinfection [4]. There is also evidence suggesting that PZQ may be having a reduced efficacy in the treatment of infection [5]. 

So far, a considerable number of schistosome antigens have been identified and tested as vaccines and, although a number of these vaccine candidates (for example, *Sm*TSP-2, *Sm*14, *Sm*29, *Sm*CB1, and *Sm*p80) have shown promising efficacy in animal models and are in various stages of pre-clinical or clinical development, none have been approved for licensure [reviewed in [3]]. 

Due to the fundamental roles they play in parasite biology (reviewed in [6]), schistosome cholinesterases (*Sm*ChEs) have been posited as intervention targets against schistosomiasis and there are several indications to support the feasibility of their use as vaccines. Firstly, *Sm*ChEs have been localized to the tegument (outer, host-exposed surface) of schistosomula and adult worms [7,8] and anti-*Sm*ChE antibodies have been shown to bind to and kill schistosomula [9], suggesting that the enzymes are accessible to immune attack. Anti-*Sm*ChE antibodies also showed no cross-reactivity against human acetylcholinesterase (AChE) [7], indicating that a vaccine safe for human use could be designed. Thirdly, protein array studies have detected significantly high levels of antibodies to *Sm*ChEs in humans exhibiting a resistance and low pathology to schistosomiasis, suggesting an involvement of these antibodies in a protective anti-schistosomal response [10,11]. Most of these studies have employed the use of anti-*Sm*ChE antibodies raised against biochemically purified material, so the vaccine efficacy of any one *Sm*ChE paralog remains to be elucidated.

A recent study by us [8] documented the existence of three *Sm*ChE paralogs (two acetylcholinesterases—*smache1* and *smache2*—and one butyrylcholinesterase (BChE)—*smbche1*), and we showed that each molecule localized to the tegument of adults and schistosomula and demonstrated, through RNAi-mediated suppression *in vitro* and *in vivo*, that each paralog was essential to parasite survival. We also reported a significant reduction in the glucose-scavenging ability of silenced parasites, providing evidence for the involvement of tegumental AChE in the mediation of exogenous glucose uptake, which has also been documented by other studies [12,13,14]. Despite the fundamental roles that SmChEs appear to play in parasitism, it remains to be determined which *Sm*ChE paralogs are effective vaccine targets in *Schistosoma mansoni*.

Herein, we demonstrate that purified IgG against each of the three *Sm*ChE paralogs inhibits ChE activity in both larval and adult worms *in vitro*, which results in eventual parasite death. Furthermore, we document the efficacy of these *Sm*ChEs, when administered as recombinant vaccines in isolation or as a triple combination, in reducing the parasite burden, stunting worm growth, and decreasing egg viability.

## 2. Materials and Methods 

### 2.1. Ethics Statement 

All experimental procedures reported in the study were approved by the James Cook University (JCU) animal ethics (Ethics approval numbers A2391). Mice were maintained in cages in the university’s quarantine facility (Q2152) for the duration of the experiments. The study protocols were in accordance with the 2007 Australian Code of Practice for the Care and Use of Animals for Scientific Purposes and the 2001 Queensland Animal Care and Protection Act.

### 2.2. Parasites 

*Biomphalaria glabrata* snails infected with *S. mansoni* (NMRI strain) were obtained from the Biomedical Research Institute (BRI) (MD, USA). Cercariae were shed by exposure to light at 28 °C for 1.5 h and mechanically transformed to obtain schistosomula [15]. To obtain adult worms, 6-8-week-old male BALB/c mice (Animal Resource Centre, WA) were infected with 120 cercariae via tail penetration and parasites harvested by vascular perfusion at 7-8 weeks post-infection [16]. 

### 2.3. Recombinant Protein Expression and Purification 

Complete open reading frames (ORFs) for *smache1*, *smbche1*, and *smache2* were synthesized by Genewiz. Attempts to express full-length sequences in *E. coli* were unsuccessful, so primer sets incorporating *Nde*I (forward primer) and *Xho*I restriction enzyme sites (reverse primer) were designed to amplify partial, non-conserved regions of each *smche* [8], which might prove more amenable to expression. Sequences (containing *Nde*I/ *Xho*I sites) for each *Sm*ChE were amplified from each full-length template by a polymerase chain reaction (PCR) and cloned into the pET41a expression vector (Novagen) such that the N-terminal GST tag was removed. Protein expression was induced for 24 h in *E. coli* BL21 (DE3) by the addition of 1 mM isopropyl beta-D-1-thiogalactopyranoside (IPTG) using standard methods. Cultures were harvested by centrifugation (8000× *g* for 20 min at 4 °C), re-suspended in 50 mL lysis buffer (50 mM sodium phosphate, pH 8.0, 300 mM NaCl, 40 mM imidazole) and stored at −80 °C. Cell pellets were lysed by three freeze-thaw cycles at −80 and 42 °C, followed by sonication on ice (10 × 5 s pulses [70% amplitude] with 30 s rest periods between each pulse) with a Qsonica Sonicator. Triton X-100 was added to each lysate at a final concentration of 3% and incubated for 1 h at 4 °C with end-over-end mixing. Insoluble material (containing *Sm*ChEs) was pelleted by centrifugation at 20,000× *g* for 20 min at 4 °C. The supernatant was discarded, and inclusion bodies (IBs) were washed twice by resuspension in 30 mL of lysis buffer, followed by centrifugation at 20,000× *g* for 20 min at 4 °C. IBs were then solubilized sequentially by resuspension in 25 mL lysis buffers containing either 2, 4, or 8 M urea; end-over-end mixing overnight at 4 °C; and centrifugation at 20,000× *g* for 20 min at 4 °C. Finally, supernatant containing solubilized IBs was diluted 1:4 in lysis buffer containing 8M urea and filtered through a 0.22 μm membrane (Millipore). Solubilized IBs were purified by immobilized metal affinity chromatography (IMAC) by loading onto a prepacked 1 mL His-Trap HP column (GE Healthcare) equilibrated with lysis buffer containing 8M urea at a flow rate of 1 mL/min using an AKTA-pure-25 FPLC (GE Healthcare). After washing with 20 mL lysis buffer containing 8M urea, bound His-tagged proteins were eluted using the same buffer with a stepwise gradient of 50-250 mM imidazole (50 mM steps). Fractions containing *Sm*ChEs (as determined by SDS-PAGE) were pooled and concentrated using Amicon Ultra-15 centrifugal devices with a 3 kDa MWCO and quantified using the Pierce BCA Protein Assay kit. The final concentration of each *Sm*ChE was adjusted to 1 mg/mL and proteins were aliquoted and stored at −80 °C.

### 2.4. Generation of Anti-Smche Antisera and Purification of IgG

Three groups of five male BALB/c mice (6-week-old) were intraperitoneally immunized with either *Sm*AChE1, *Sm*BChE1, or *Sm*AChE2 subunits (50 μg/mouse). Antigens were mixed with an equal volume of Imject alum adjuvant (Thermofisher) and administered three times, two weeks apart. Two weeks after the final immunization, mice were sacrificed and blood was collected via cardiac puncture. Blood from all mice in each group was pooled and serum was separated by centrifugation after clotting and stored at −20 °C. Polyclonal antibodies were purified from mouse sera using Protein A Sepharose-4B (Thermofisher), according to the manufacturer’s instructions. Serum from naïve mice was similarly processed.

### 2.5. Effect of Polyclonal Anti-SmChE IgG on Larval Worms 

Newly transformed schistosomula (1000/mL) were cultured in Dulbecco’s Modified Eagle Medium (DMEM) (supplemented with 4 × AA) at 37 °C and 5% CO_2_ in the presence of 50 µg of either anti-*Sm*AChE1, *Sm*BChE1, or *Sm*AChE2 polyclonal IgG or a combination of all three antibodies (equal amounts—50 µg total). Separate sets of parasites were similarly incubated with 50 µg of naïve mouse IgG, which served as a control. After 2 and 14 h (separate experiments were conducted for each timepoint), 300 schistosomula from each experiment were removed and assessed for viability using Trypan Blue exclusion (100 parasites in triplicate) [17]. Surface and secreted ChE activity was measured using the remaining parasites by incubating them in 0.5 mL of assay buffer (0.1M sodium phosphate, pH 7.4, 2 mM acetylthiocholine [AcSCh], or 2 mM butyrylthiocholine [BcSCh], and 0.5 mM 5, 5′-dithio-bis 2-nitrobenzoc acid [DTNB]) and monitoring the absorbance increase (AcSCh conversion or BcSCh conversion) over 1 h at 405 nm in a Polarstar Omega microplate reader (BMG Labtech). Parasites cultured with naïve IgG served as a negative control. Data are presented as the average of two biological and three technical replicates ± SEM.

### 2.6. Effect of Polyclonal Anti-SmChE IgG on Adult Worms 

Enzyme inhibitory effects 24 h after the addition of IgG (including naïve IgG) were measured as for schistosomula using five pairs of freshly perfused adult worms in 1 mL of media. Data are presented as the average of two biological and two technical (four total) replicates ± SEM. To investigate the effects of polyclonal anti-*Sm*ChE IgG on the worm viability, ten pairs of worms were similarly incubated with antibodies for 10 days, monitored every 24 h for motility by microscopic examination, and considered dead if no movement was seen. To measure the effect of polyclonal anti-*Sm*ChE IgG on glucose uptake, ten pairs of worms were similarly incubated with antibodies for 24 h and transferred to DMEM (1000 mg/L) and then the media glucose concentration was measured over a 24 h period using a glucose assay kit (Sigma). Glucose levels were expressed relative to media collected from worms which received naïve IgG (negative control). Data are presented as the average of two biological replicates ± SEM. 

### 2.7. Anti-SmChE IgG Responses in S. mansoni-Infected Mice during Infection and Before and After PZQ Treatment 

Sera from *S. mansoni*-infected male BALB/c mice (6-8 weeks) (n = 5) were collected at day 3, 14, 28, 42, and 56 post infection (p.i.) to assess anti-*Sm*ChE responses during the course of parasite infection. In a separate experiment, sera from *S. mansoni*-infected male BALB/c mice (6-8 weeks) (n = 11) were collected at 5 weeks p.i. and mice were then treated orally with PZQ (100 mg/kg) at 35, 37, and 39 days p.i. Sera were again collected at day 49 p.i. (2 weeks post-PZQ treatment). Anti-*Sm*ChE responses during infection and before and after PZQ treatment were screened by ELISA with plated *Sm*ChEs (100 ng/well) using standard methods. The cutoff value for each dilution was established as three times the mean OD of the naïve sera for that dilution and the endpoint was defined as the highest dilution above the cutoff value. 

### 2.8. Vaccine Trials

Five groups of 10 male BALB/c mice (6-8 weeks) were immunized intraperitoneally on day 1 (50 μg/mouse) with either *Sm*AChE1, *Sm*BChE1, *Sm*AChE2, a combination of all three *Sm*ChEs (17 μg each—50 μg total), or PBS, each formulated with an equal volume of Imject alum adjuvant (Thermofisher) and 5 μg of CpG ODN1826 (InvivoGen). Immunizations were repeated on day 15 and 29 and each mouse was infected by tail penetration with 120 *S. mansoni* cercariae [18] on day 43. Two independent trials were performed to ensure reproducibility. Blood was sampled at day 28 and 42 and on the day of a necropsy, to determine pre- and post-challenge antibody titers. 

### 2.9. Mouse Necropsy and Estimation of Worm and Egg Burden

Mice were necropsied at day 91 (7 weeks p.i.) and worms were harvested by vascular perfusion and counted. Worms from the mice in each group were pooled and a random sample of each pool was photographed and measured using ImageJ software. Livers were removed and halved, with one half weighed and digested for 5 h with 5% KOH at 37 °C with shaking. Schistosome eggs from digested livers were concentrated by centrifugation at 1000× *g* for 10 min and re-suspended in 1 mL of 10% formalin. The number of eggs in a 5 μL aliquot was counted in triplicate and the number of eggs per gram (EPG) of the liver was calculated. Small intestines were removed and cleaned of debris before being weighed and digested as per the liver halves. Eggs were also similarly concentrated and counted to calculate the intestinal EPG.

### 2.10. Egg Viability Assays

The other half of each liver was pooled according to the group, homogenized in H_2_O, and placed in identical foil-covered volumetric flasks under bright light to hatch eggs released from the livers. After 1 h, the number of miracidia in 10 × 50 μL aliquots of H_2_O (sampled from the extreme top of each flask) were counted. The number of eggs in each flask at the start of the hatching experiment was determined by liver EPG calculations, allowing the egg hatching index of each group to be calculated by expressing the hatched eggs (miracidia) as a percentage of the total eggs [13].

### 2.11. Glucose Consumption and Glycogen Storage Assays

Five pairs of freshly perfused worms from each vaccinated group were cultured in DMEM (1000 mg/L glucose). Media (50 µL) from each experiment was collected after 24 h, and the amount of glucose was quantified using a colorimetric glucose assay kit (Sigma), according to the manufacturer’s instructions. Glucose levels were expressed relative to media collected from worms recovered from PBS-treated mice (negative control). To measure the glycogen content of these worms, Triton X-100-soluble extracts of each group of five pairs of worms (made by homogenizing the parasites in 1% Triton X-100, 40 mM Tris-HCl, pH 7.4, mixing overnight at 4 °C, and collecting the supernatant by centrifugation at 15,000× *g* for 1 h at 4 °C) were assayed for glycogen in a modified procedure described by Gomez-Lechon et al. [19]. Briefly, 0.2 M sodium acetate, pH 4.8, was added to 30 μg parasite extract and 50 μL glucoamylase (10 U/mL) to make a reaction volume of 150 μL. The mixture was incubated at 40 °C for 2 h with shaking at 100 rpm, 40 μL was added to a new microplate with 10 μL 0.25 M NaOH, and the amount of glucose was quantified using the colorimetric glucose assay kit. Only worms from trial 1 were available for these experiments. Extracts were made from triplicate sets of parasites and assays were performed three times. Data are presented as the average of each triplicate biological and technical experiment ± SEM. 

### 2.12. Experiments Involving Sera from Vaccinated Mice

Sera were collected from all mice in each group before cercarial challenge and at necropsy. Serum anti-*Sm*ChE IgG titers were measured by ELISA against plated *Sm*ChEs using standard methods. The cutoff value for each dilution was established as three times the mean OD of the naïve sera for that dilution and the endpoint was defined as the highest dilution above the cutoff value.

To assess whether vaccination-induced antibodies would interact with host serum AChE or BChE, the AChE and BChE activity of pre-challenge sera from all vaccinated and control mice was measured by the Ellman assay. Briefly, 1 μL of pre-challenge serum from each mouse was added to 200 μL assay buffer (0.1M sodium phosphate, pH 7.4, 2 mM acetylthiocholine [AcSCh], or 2 mM butyrylthiocholine [BcSCh], and 0.5 mM 5, 5′-dithio-bis 2-nitrobenzoc acid [DTNB]) and the absorbance increase (AcSCh conversion or BcSCh conversion) was monitored over 20 min at 405 nm in a Polarstar Omega microplate reader (BMG Labtech). Data are presented as the average of three technical replicates ± SEM.

### 2.13. Statistical analyses

Statistical differences for all experiments in this chapter were calculated by the Student’s *t* test using GraphPad Prism 7 software. Results are expressed as the mean ± standard error of the mean (SEM).

## 3. Results

### 3.1. Anti-SmChE Polyclonal Antibodies Block Enzyme Activity and Decrease the Viability of Larval S. mansoni in Vitro

To determine the ability of anti-*Sm*ChE-specific polyclonal antibodies to inhibit ChE activity in *S. mansoni*, and the effect this had on parasite viability, we studied the effects of paralog-specific antibodies on schistosomula at two different timepoints. Treating schistosomula with anti-*Sm*AChE1 IgG, anti-*Sm*AChE2 IgG, or a cocktail of all three anti-*Sm*ChE IgGs caused a significant inhibition (*p* ≤ 0.01) of AChE activity of 56.2%, 57.1%, and 59.74%, respectively, 2 h after treatment, in comparison with the naïve IgG control (Figure 1A). When schistosomula were incubated with anti-*Sm*BChE1 IgG or a cocktail of all three anti-*Sm*ChE IgGs for 2 h, the BChE activity was inhibited by 37.4% (*p* ≤ 0.01) and 49.3% (*p* ≤ 0.001), respectively, compared to the control (Figure 1B). Schistosomula viability was not significantly affected at this timepoint (Figure 1C). Extending the treatment with anti-*Sm*AChE1 IgG, anti-*Sm*AChE2 IgG, or a cocktail of all three anti-*Sm*ChE IgGs for 14 h significantly decreased (*p* ≤ 0.001) the AChE activity by 66.9%, 70.5%, and 72.6%, respectively, compared to the control (Figure 1D). Similarly, when schistosomula were incubated with anti-*Sm*BChE1 IgG or a cocktail of all three anti-*Sm*ChE IgGs for 14 h, the BChE activity decreased significantly (*p* ≤ 0.01) by 26.5% and 35.6%, respectively, compared to the control (Figure 1E). Schistosomula viability was significantly decreased (*p* ≤ 0.01) by all treatments at this timepoint, with the biggest decrease seen in the anti-*Sm*ChE cocktail IgG-treated group (Figure 1F).

### 3.2. Effects of Anti-SmChE Antibodies on Adult Worms

The effects of anti-*Sm*ChE antibodies on *S. mansoni* adult worms was also tested. Freshly perfused worms cultured in the presence of anti-*Sm*AChE1, anti-*Sm*AChE3, or a cocktail of all three anti-*Sm*ChEs, showed no significant inhibition of AChE or BChE activity at 2 h post-treatment, compared to controls. After 24 h treatment, however, all anti-*Sm*ChE IgG-treated groups showed a significant inhibition of AChE and BChE activity, with the anti-*Sm*ChE cocktail IgG-treated group displaying the greatest inhibition of AChE activity (Figure 2A,B). The rate of glucose uptake over 24 h was also measured at this timepoint and all antibody treatments significantly reduced the glucose uptake in adult worms, compared with naïve IgG-treated controls, again with the anti-*Sm*ChE cocktail IgG-treated group displaying the greatest inhibition (Figure 2C). To determine if anti-*Sm*ChE antibodies can play a role in killing adult worms, the antibody experiment was repeated with ten pairs of adult worms per treatment and worm viability post-treatment was assessed. Consistent with inhibition of AChE activity and glucose uptake, the cocktail of anti-*Sm*ChE antibodies was the most effective at killing (all worms dead at day 7 post-treatment), compared to controls (Figure 2D). 

### 3.3. Antibody Responses to SmChEs during the Course of Infection and Following PZQ Treatment in Mice

Antibody responses to all *Sm*ChEs were significantly higher in infected mice than before infection and increased as infection progressed (Figure 3A). In a separate experiment, all anti-*Sm*ChE IgG responses were shown to significantly increase (*p* ≤ 0.001) after PZQ treatment (Figure 3B).

### 3.4. Vaccine Efficacy of Recombinant SmChEs in a Mouse Model of Schistosomiasis

In both vaccine trials, all four groups of mice immunized with *Sm*ChEs, either in isolation or as a combination, showed a significant decrease in worm burden (28–39%), compared to controls, with the *Sm*ChE cocktail-vaccinated group displaying the highest reduction in trial 1 and 2 of 39% (*p* ≤ 0.0009) and 38% (*p* ≤ 0.0001), respectively (Figure 4A,B). Compared to controls, significant decreases in liver egg burdens (expressed as eggs per gram—EPG) were observed for all groups across both trials (13–46%), except for the *Sm*AChE1-vaccinated group in trial 1. When averaged over both trials, liver egg burdens in the cocktail-vaccinated group showed the greatest reduction (Figure 4C,D). Intestinal egg burdens (expressed as EPG—only determined for trial 2) were significantly reduced for all vaccinated groups, compared to controls, with the greatest reduction seen in the group vaccinated with the *Sm*ChE cocktail (33%, *p* ≤ 0.001). (Figure 4E). Egg viability (only assessed for trial 2), as determined by egg hatching from liver homogenates, was significantly reduced in the groups vaccinated with the *Sm*ChE cocktail (40%, *p* ≤ 0.01) and *Sm*AChE2 (46%, *p* ≤ 0.01) (Figure 4F). While there was no significant reduction in glucose uptake for worms from any of the vaccinated groups, compared to controls, the glycogen content of worms from all vaccinated groups was significantly lower (24–52%, *p* ≤ 0.001) than worms from the control group (Figure 5A). A significant reduction in worm length (30%-50%) was also observed between worms from all vaccinated groups compared to worms from the control group (Figure 5B). As with the parasitology burden data, worms from the cocktail-vaccinated group showed the greatest decrease in glycogen content and body length. The glucose uptake, glycogen content, and worm size were not significantly different between control and vaccinated groups in trial 2. Serum AChE and BChE activity was also not significantly different between control and vaccinated mice (only measured for trial 2) (Appendix A).

### 3.5. Antibody Responses in Vaccinated Mice

Moderate to high (>1,000,000) anti-*Sm*ChE endpoint titers were seen in the pre-challenge serum of all mice in all vaccinated groups. Post-challenge titers in all groups were four- to ten-fold lower than pre-challenge titers (Appendix A). There was no correlation between pre-challenge titers and worm burdens in any groups.

## 4. Discussion

Surface-exposed proteins and secreted proteins are effective targets for vaccine development in schistosomes due to their capacity for interaction with host antibodies [20,21]. In this regard, *Sm*ChEs are promising candidates, as we have shown in previous immunolocalization studies of *S. mansoni* that *Sm*AChE1, *Sm*BChE1, and *Sm*AChE2 are expressed in the tegument of adult worms and schistosomula and proteomic analysis of *S. mansoni* excretory/secretory (ES) products has confirmed the presence of *Sm*AChE1 and *Sm*BChE1 [8]. Furthermore, RNAi-mediated silencing of all three *Sm*ChE genes, both individually and in combination, significantly decreased the schistosomula viability *in vitro* and parasite survival in vivo [8], implying that these genes are essential for proper worm development and function. Moreover, recent protein array studies have demonstrated high levels of circulating antibodies to *Sm*BChE1 in individuals exhibiting drug-induced resistance and a low pathology reaction to schistosomiasis, implicating these antibodies in a protective anti-schistosomal response [10,11]. 

Given that *Sm*ChEs are accessible to antibody attack and are enzymatically functional [8], catalytic activity can be used to measure the effectiveness of antibody binding as the interaction between enzymes and their corresponding antibodies generally leads to a complete or partial reduction in their enzymatic activity [22,23,24]. The data presented herein show that antibodies against recombinant *Sm*ChEs are capable of inhibiting surface (and, in the case of *Sm*AChE1 and *Sm*BChE1, secreted) enzymatic activities in both schistosomula and adult worms, which is similar to previous studies that have used antibodies raised against parasite-derived AChE to inhibit AChE activity on intact *S. mansoni* [25], *Necator americanus* [26], *Dictyocaulus viviparus* [27], and *Electrophorus electricus* [28]. The inhibitory effect of antibodies is potentially due to steric hindrance, potentially blocking substrate access to the peripheral anionic site or catalytic gorge of AChE. Indeed, previous studies on rabbit [29], human [30], and bovine [31] AChE have documented the AChE-inhibitory ability of antibodies raised against epitopes other than AChE-active sites.

Consistent with the effects of RNAi-mediated *smche* gene silencing [8], antibody-mediated *Sm*ChE inhibition resulted in a significant decrease in parasite viability. We posit that this mode of enzyme inhibition was the cause of eventual parasite death as schistosomula were still viable, compared to controls, at 2 h after antibody treatment, despite a significant decrease in ChE activity; it was not until after a much longer exposure (14 h) to anti-*Sm*ChE antibodies that the parasite viability was significantly lower.

Glucose uptake in adult worms was also significantly reduced by anti-*Sm*ChE antibody treatment. The cholinergic action of surface AChE has been implicated in mediation of the glucose scavenging mechanism in schistosomes [12], AChE-inhibitory metal complexes reduce glucose import in the parasites [13], and we and others have shown that RNAi-mediated silencing of schistosome *che* genes lessens the uptake of glucose by these parasites [8,14], so it is possible that antibody-mediated impairment of AChE involvement in the glucose uptake pathway is the cause of this effect. It may be that there is some redundancy in the cholinergic functioning of these molecules (even BChEs, like *Sm*BChE1, can perform a cholinergic role in situations of AChE deficiency [32]), so collective inhibition of the molecules is required to produce a functional deficit. Additionally, given the multiple proposed functions for parasite ChEs [6,33], it is possible that the neutralization of multiple enzymatic targets more profoundly interrupts varied processes of parasite biology than just cholinergic transmission. 

Given the relative cytotoxic potential of antibodies against all three *Sm*ChEs, as opposed to any single *Sm*ChE, we decided to test the efficacy of this antigen cocktail as a vaccine in a mouse model of schistosomiasis. The vaccine efficacy of each individual *Sm*ChE was also tested to investigate the relative anti-parasitic effects of each *Sm*ChE over the cocktail or one another.

Mice vaccinated with the cocktail of *Sm*ChE antigens displayed the highest level of protection against experimental schistosomiasis, showing the greatest reductions in every parameter tested. An additive protective effect was not readily apparent, however, as protection levels were not significantly different from groups vaccinated with single antigens. Of the groups vaccinated with individual *Sm*AChEs, the *Sm*AChE2-vaccinated group engendered the highest levels of protection. Similar results were reported in a test on the vaccine efficacy of a recombinant AChE from *Schistosoma japonicum* [14]. Furthermore, vaccine trials using purified secretory AChE from the nematodes *Trichostrongylus colubriformis* and *Dictyocaulus viviparous* have resulted in significant protection in animal models [34,35]. It should be noted that schistosomes and other helminth parasites have complex life cycles involving more than one host, so rely on passage of their eggs from the definitive host into the environment to continue their life cycle and transmit disease. Accordingly, mathematical modeling supports the view that a vaccine inducing even partial protection would decrease the parasite egg load in the environment, contributing to the reduction of schistosome infections and interrupting transmission in endemic areas [1,36].

Egg burdens did not concomitantly decrease with worm burdens, but there were significant reductions in egg viability in all but the *Sm*AChE1-vaccinated group, which is an observation we have previously reported when testing the in vivo anti-schistosomal efficacy of AChE-inhibitory drugs [13]. Studies in rats and honey bees have observed abdominal spasms and involuntary muscle contractions when AChE inhibitors have been administered to these organisms [37,38], so a possible explanation for this “less than expected” decrease in egg number but significant reduction in viability could be that ova are being prematurely released as a result of antibody-mediated AChE inhibition affecting reproductive tract motility. It could also be that ChE vaccination affected the fecundity and egg maturity given the significantly smaller size of worms recovered from vaccinated groups, compared to controls. Indeed, previous studies on insects demonstrated that the suppression of AChE expression considerably reduced the weight and length of surviving organisms [39,40,41] and severely affected the hatching ability of the eggs laid [39,42]. These reports have suggested that dysregulated cell proliferation and apoptosis during larval growth may be reasons for such phenotypic effects attributed to the absence of AChE, although such a link in trematodes remains to be established. Finally, parasites recovered from vaccinated mice had significantly depleted glycogen stores. Reduced glycogen content and glucose uptake have been previously observed in worms treated with AChE-inhibitory drugs *in vitro* [13] and attributed to interference with the tegumental AChE-mediated glucose scavenging pathway [12] through the inhibition of this enzyme. It could be that the same effect is being orchestrated by the antibody-mediated inhibition of AChE (which would be consistent with the results of *in vitro* antibody-based experiments), forcing the parasite to rely on its glycogen stores, rather than the scavenging of exogenous glucose, for nutrition.

Even though immunization with *Sm*ChEs induced high antibody titers, these antibody levels were not sustained during the course of infection, with titers at necropsy dropping between four- and ten-fold from pre-challenge levels. This would seem to indicate that the specific antibody response induced by immunization was not augmented by natural infection, which is a hypothesis corroborated by the generation of modest anti-*Sm*ChE titers during the course of parasite infection in a separate experiment. That being said, the *Sm*ChEs used in this study were still capable of inducing moderate levels of protection in the face of modest antibody titers. Treatment with PZQ has been shown to induce antibody-mediated resistance to schistosomiasis in humans through the exposure of parasite antigens to the immune system (and subsequent generation of an antibody response) as a result of tegument damage [43] and protein array studies conducted by us have shown that antibodies to *Sm*BChE1 are significantly upregulated in resistant individuals [10]. The upregulation of *Sm*ChE immune responses following PZQ treatment has been verified in this study. Given that an effective anti-schistosomal vaccine strategy would ideally be linked with chemotherapy [10], it is possible that the vaccine efficacy of antigens such as the ones described here could be increased by vaccination after PZQ treatment due to the augmentation of an already upregulated immune response. The current World Health Organisation (WHO) objective regarding schistosomiasis is to eliminate disease morbidity and mortality as a public health problem—defined by reaching a prevalence of ≤5% and ≤1% of heavy-intensity infections, respectively, in school-age children—in the coming decade [44]. Vaccine-linked chemotherapy has been deemed one the most effective control measures for combating the disease, with this strategy combining the mainstay of schistosomiasis intervention (mass administration of PZQ) with the more long-term benefits of vaccination, and recent modeling has predicted that chemotherapy linked with a vaccine that has an effective duration of protection is the best intervention strategy for achievement of the WHO’s goals to significantly reduce the burden of disease caused by schistosomiasis [44].

## Figures and Tables

**Figure 1 vaccines-08-00162-f001:**
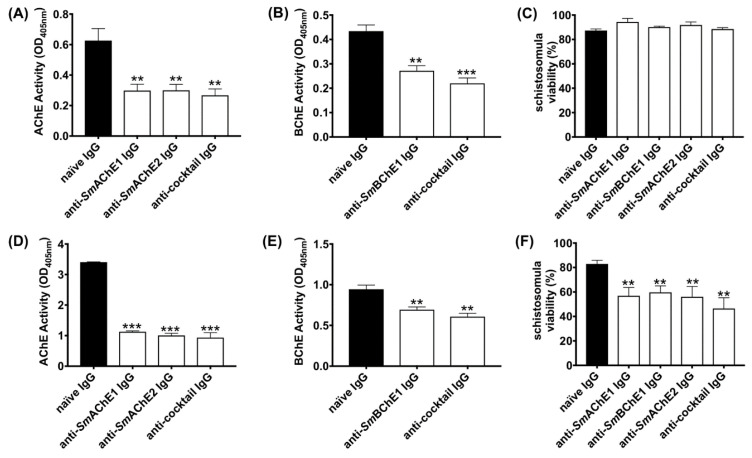
Anti-schistosome cholinesterase (*Sm*ChE) antibodies inhibit ChE activity in schistosomula, which leads to a decreased parasite viability. Newly transformed schistosomula (1000/treatment) were incubated in Dulbecco’s Modified Eagle Medium (DMEM) in the presence of anti-*Sm*ChE IgG and incubated at 37 °C in 5% CO_2_. Naïve IgG served as a negative control. (**A**) AChE activity 2 h after treatment. (**B**) BChE activity 2 h after treatment. (**C**) Schistosomula viability 2 h after treatment. (**D**) AChE activity 14 h after treatment. (**E**) BChE activity 14 h after treatment. (**F**) Schistosomula viability 14 h after treatment. Data represents the mean ± SEM of two biological and three technical replicates. Significance (relative to the naïve IgG control) determined by the student’s *t* test, where ** *p* ≤ 0.01 and *** *p* ≤ 0.001.

**Figure 2 vaccines-08-00162-f002:**
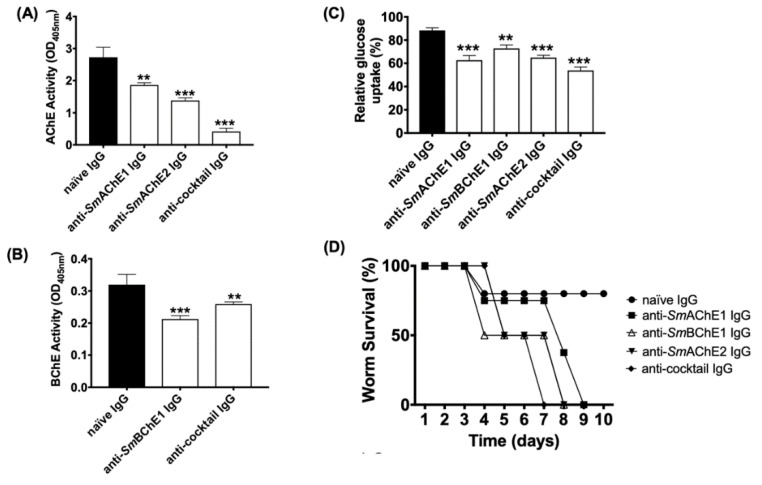
Effects of anti-*Sm*ChE antibodies on adult *Schistosoma mansoni* worms. Ten pairs of freshly perfused worms were incubated in the presence of anti-*Sm*ChE purified IgG in DMEM at 37 °C and 5% CO_2_. Naïve IgG served as a negative control. (**A**) AChE activity 24 h after treatment. (**B**) BChE activity 24 h after treatment. (**C**) Glucose uptake over 24 h, one day after treatment. (**D**) Survivability up to 10 days after treatment. The results are the mean ± SEM of two biological and three technical replicates (**A**–**C**) or two biological replicates (D). Significance (relative to the naïve IgG control) determined by the student’s *t* test, where * *p* ≤ 0.05, ** *p* ≤ 0.01, and *** *p ≤* 0.001.

**Figure 3 vaccines-08-00162-f003:**
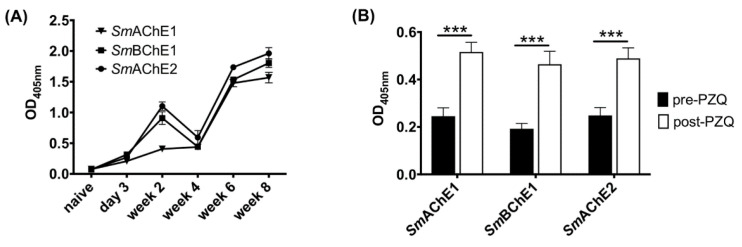
Antibody responses to *Sm*ChEs during the course of infection and following praziquantel (PZQ) treatment in mice. ELISAs showing anti-*Sm*ChE IgG responses in mice (**A**) from 3 days to 8 weeks post-infection (*n* = 5) and (**B**) before (5 weeks post-infection) and 2 weeks after PZQ treatment (*n* = 11). Data represents the mean of two technical replicates and significance was determined by the student’s *t* test, where *** *p* ≤ 0.001.

**Figure 4 vaccines-08-00162-f004:**
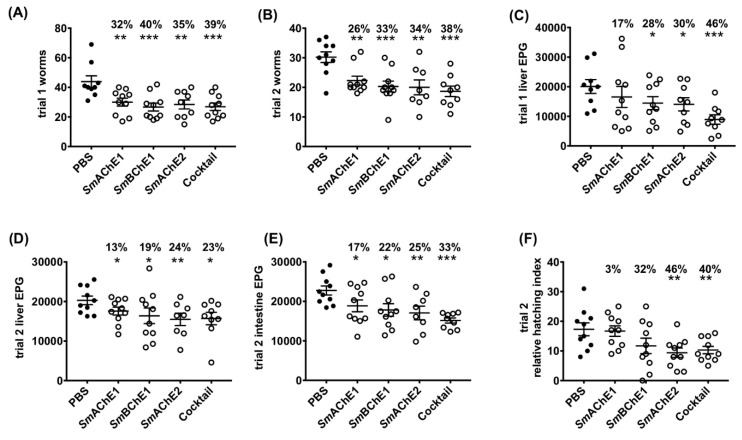
Vaccine efficacy of recombinant *Sm*ChEs in a mouse model of schistosomiasis. Graphs show parasitology burdens from vaccinated and control mice. (**A**) Trial 1 adult worms. (**B**) Trial 2 adult worms. (**C**) Trial 1 liver eggs per gram (EPG). (**D**) Trial 2 liver EPG. (**E**) Trial 2 intestinal EPG. (**F**) Hatching viability of eggs obtained from the pooled livers of control and vaccinated mice from trial 2. Data are the average of ten replicate counts ± SEM of hatched miracidia. Significance and percent reductions (if any) for all parameters are measured relative to the control group. Significance determined by the student’s *t* test, where * *p* ≤ 0.05, ** *p* ≤ 0.01, *** *p* ≤ 0.001.

**Figure 5 vaccines-08-00162-f005:**
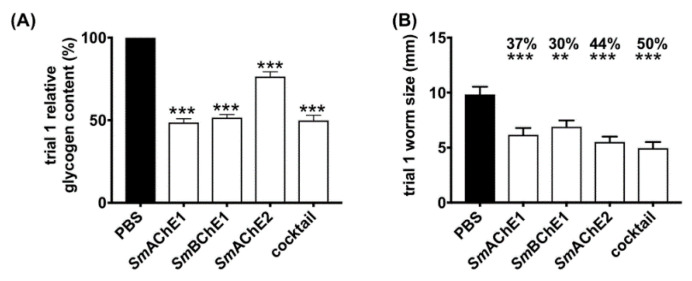
Effect of *Sm*ChE vaccination on glycogen storage in, and the size of, *S. mansoni* adult worms. (**A**) Triton-X-100 extracts were made from five pairs of worms freshly perfused from each vaccinated or control group and the glycogen content in these extracts was measured. Plotted data are the average of triplicate biological and technical experiments ± SEM. (**B**) Worm sizes (mm) were assessed by randomly selecting and measuring (ImageJ) at least 20 worms from each group. Significance and percent reductions (if any) for both parameters were measured relative to the control group. Differences for both experiments were measured by the student’s *t* test, where ** *p* ≤ 0.01 and *** *p* ≤ 0.001.

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
