# Peer review of "Vaccination with Schistosoma mansoni Cholinesterases Reduces the Parasite Burden and Egg Viability in a Mouse Model of Schistosomiasis"

_vaccines, 2020, doi:10.3390/vaccines8020162_

Round 1

Reviewer 1 Report

This is a well written paper with well presented results. To this reviewer there is one weakness that dampens my encouragement for publication of the paper and its conclusions in its current state and that is related to the fact that the waiting time between the second vaccine challenge and the parasite injection is only 14 days. That means the parasite enter the system right at the peak of the second immune activation and that is not how a vaccine is supposed to work. One should wait for the immune response to return to 'normal' (like it would be the case in a natural habitat) and only later challenge with parasite to see if you can trigger an vaccine recall response. This is crucial because most 'promising' vaccine candidates get lost in the process because they fail to establish a memory cell population that can be efficiently triggered months later. To have a promising candidate, one should wait at least 8 week between the second boost and the actual parasite challenge.

Author Response

We thank reviewer 1 for making this important observation about the time between final immunisation and parasite challenge. It is standard practice for helminth vaccine trials to be designed with approximately 14 days between these two events and there is extensive literature (reported by us and others) documenting the efficacy of schistosomiasis and hookworm vaccines when trialled in this way (EG: Tran, Pearson et al., Nature Med, 2006; Pearson et al., FASEB J, 2009; Pearson et al., PLoS Negl Trop Dis, 2012; Mossallam et al., BMC Infect Dis, 2015; You et al., Int J Mol Sci, 2018), including schistosomiasis and hookworm vaccine candidates which have progressed into human clinical trials (Merrifield et al., Vaccine, 2016; Hotez et al., Vaccine, 2016). Further, thi vaccination strategy is relevant to a natural scenario as people would be vaccinated in areas of constant schistosomiasis transmission and, therefore, would be regularly exposed immediately after vaccination.

Reviewer 2 Report

Schistosomiasis, considered one of the neglected tropical diseases by the WHO, remains an important public health problem throughout many resource-poor regions of the world. Despite decades of research there is still no effective or approved vaccine for this helminthic infection. In this communication, the authors have postulated that schistosome cholinesterases (SmChEs), located on the tegument of schistosomulas and adult worms, may be suitable targets for effective vaccines.

In this well-written and well-organized original investigation the authors prepared purified IgG against each of 3 SmChE paralogues from intraperitoneal inoculation of mice, and these antibodies were tested against schistosomula which were evaluated for viability. Adult schistosomes were also tested for viability following exposure. Mice were challenged with either SmAChE1, SmBChE1 or SmAChE2, or all 3, necropsied, and the Schistosoma mansoni egg burden quantitated. Assays for ova viability, glucose and glycogen storage were also performed. The authors also evaluated whether vaccination-induced antibodies would interact with host serum SmChE activity in re-challenge sera. Statistical analyses of data were performed using the appropriate standard methods of testing. The results were very interesting and provided evidence that SmChEs are promising candidates as effective targets for vaccine development.

The illustrative diagrams and charts in this communication are well-prepared, clearly understood, highlight the text and results, and are complimentary with the figure legends.

In summary, this is a potentially significant contribution to the literature on vaccine preparation for Schistosoma mansoni in which the authors have convincingly demonstrated that vaccination with schistosome cholinesterases can result in a reduction of parasite burden and egg viability in the murine model of infection.

The references are through and up-to-date... however; there are a number of formatting inconsistences in the Reference section which will need to be revised, for example, capitalization of journal titles (#11,13), mixture of journal abbreviations and full journal titles, irregular left margin (#30), capitalization of titles of articles (#5), etc.

Author Response

We would like to thank reviewer 2 for their positive appraisal of the manuscript. In accordance with the review, formatting inconsistencies within the reference list have been corrected and the revised ms is attached. 

Round 2

Reviewer 1 Report

Thank you for the answer. I guess that is the difference between a fundamental immunologist and a field oriented immunologist. If it works in the given context, I am OK with it.